

# Denoising and enhancement for robust image steganography

Dapeng Cheng, Yue Kong, Yanyan Mao and Liunian Bian

School of Computer Science and Technology, Shandong Technology and Business University, Yantai City, Shandong Province, China

## ABSTRACT

Image steganography aims to embed secret information into a cover image in such a manner that the hidden content remains visually imperceptible while still being accurately recoverable when needed. However, traditional image steganography methods often suffer from limited robustness and are highly susceptible to common image distortions such as Gaussian noise, Poisson noise, and lossy compression. To address these limitations, this article proposes DERIS, a robust image steganography model based on invertible neural networks (INNs), which enhances resistance to image distortions through structural design. The model integrates identical denoising enhancement modules both before the discrete wavelet transform (DWT) and after the inverse discrete wavelet transform (IDWT) in the backward extraction pathway, significantly improving the quality of the extracted secret images. Furthermore, a training strategy that incorporates denoising enhancement is employed to ensure the model's stability and reversibility under various types of image interference. Extensive experiments were conducted primarily on the DIV2K, ImageNet, and COCO datasets, using evaluation metrics including Peak Signal-to-Noise Ratio (PSNR), Structural Similarity Index (SSIM), Root Mean Square Error (RMSE), Mean Absolute Error (MAE), Learned Perceptual Image Patch Similarity (LPIPS), and Normalized Cross-correlation (NCC). Experimental results demonstrate that under Gaussian noise ($\sigma = 10$), the proposed method achieves a PSNR of 32.43 dB between cover and container images, and 30.24 dB between secret and extracted images on the DIV2K dataset, representing an improvement of 1.08 dB over Pris. Under JPEG compression (QF = 80), the method achieves PSNR values of 28.63 dB (cover-container) and 27.74 dB (secret-extracted) on the ImageNet dataset, which are 2.11 dB higher than those of Pris. Similarly, on the COCO dataset under the same attack condition, the method achieves PSNR values of 28.44 dB (cover-container) and 26.81 dB (secret-extracted), showing improvements of 0.91 dB over Pris. These results significantly outperform those of current state-of-the-art methods, demonstrating the enhanced robustness and practicality of the proposed approach.

Corresponding author
Dapeng Cheng,
chengdapeng@sdtbu.edu.cn

## INTRODUCTION

Image steganography is a technique for hiding secret information within a host medium, with the goal of making the altered medium visually indistinguishable from the original while allowing only specific recipients to extract the hidden information

(*Wani & Sultan, 2023*; *Sahu & Sahu, 2020*; *Zhao, Yao & Xue, 2025*; *Chen, Li & Zhao, 2023*; *Liu, Tang & Zheng, 2022*; *Megías, Mazurczyk & Kuribayashi, 2021*). Compared to traditional image encryption methods, image steganography offers a significant security advantage; the container image embedded with secret information appears identical to the original. Due to its stealth and practicality, image steganography has broad applications across multiple fields. In information security (*Xu et al., 2022*), it can conceal authentication data or encryption keys in internet or cloud environments. In digital copyright protection (*Çiftci & Sümer, 2022*), it can safeguard the intellectual property of e-books, image libraries, digital artworks, and more. In covert communication (*Li et al., 2024*), it can be used to transmit sensitive diplomatic documents or intelligence internationally. In the medical field (*Venugopalan, Gupta & Kumar, 2024*), it can enhance patient data security in telemedicine by embedding auxiliary diagnostic information into medical images, improving diagnostic efficiency. In the military domain (*Yu et al., 2023*), it can embed operational plans or instructions into satellite imagery, or integrate authentication data into military images to ensure their authenticity and prevent forgery or tampering.

Existing steganographic techniques are primarily divided into traditional steganography and deep learning-based steganography (*Hammad et al., 2025*). Traditional steganography can typically hide only a small amount of information (0.2 to 4 bits per pixel), making it difficult to meet the demands of large-capacity hiding. A representative method is LSB (Least Significant Bit) steganography, which embeds information by modifying the least significant bits of image pixels (*Kheddar & Megías, 2022*), offering the advantages of simplicity and high concealment but being vulnerable to statistical analysis, image processing, and compression attacks, resulting in low security and robustness. In recent years, deep learning techniques have been introduced into image steganography, leveraging neural networks to simultaneously perform embedding and extraction processes, significantly improving the capacity and accuracy of information hiding. For example, *Baluja (2017)* proposed the first convolutional neural network (CNN) (*Liu, Wang & Chen, 2022b*) to address image hiding, and in the same year, *Hayes & Danezis (2017)* introduced adversarial training for steganography. *Jing et al. (2021)* and others proposed the Hi-Net framework based on invertible neural networks (INNs), achieving full reversibility in the steganography and information recovery process, which allows the network to be trained once to accomplish both embedding and extraction, addressing issues related to capacity, imperceptibility, and security.

In this study, we choose images as the carrier for steganographic messages primarily because images offer larger embedding capacity and richer feature space, allowing more information to be concealed while maintaining high imperceptibility. In image steganography, payload refers to the secret information (such as a message, file, or other data) that is embedded into the cover image. The goal is to embed this information in a way that the cover image remains visually indistinguishable from the original. The size of the payload plays a critical role in steganography, as larger payloads can degrade the visual quality of the image and reduce imperceptibility. Therefore, an optimal balance must be achieved between embedding capacity and the image's perceptual quality. Payload size is

typically measured in bits per pixel (bpp), and a larger payload allows more information to be hidden but may make the embedded data more detectable. In contrast, although text can also serve as a steganographic medium, its information capacity is limited and it is more easily detected through statistical analysis or natural language processing techniques, which reduces the security of steganographic transmission. Furthermore, image steganography demonstrates superior robustness, as embedded information can still be preserved after compression, cropping, or filtering, whereas text is more susceptible to loss during processing. More importantly, image steganography can exploit high-dimensional features and subtle noise patterns to enhance resistance to detection, thereby fulfilling the core objective of steganography: fully covert communication.

Although image steganography has advantages in capacity, robustness, and detectability resistance, it fundamentally differs from digital watermarking in terms of its core goal. Watermarking is primarily used for copyright protection or information tracing, and the embedded information is expected to be detectable and recoverable even under interference. In contrast, steganography emphasizes covert communication, completely hiding the existence of secret information so that it can be transmitted without being noticed. As noted in *Rustad et al. (2023)*, the evaluation of steganographic methods focuses on imperceptibility and resistance to detection rather than traceability or copyright marking. Therefore, even when images are chosen as the message carrier to enhance robustness and concealment, this study still adheres to the core objective of steganography—ensuring the secrecy of information rather than copyright protection or traceability.

In summary, the core research focus of steganography lies in the secure transmission of secret information, embedding it into a carrier without attracting attention. Specifically, research priorities include imperceptibility, security and resistance to steganalysis, capacity, and robustness. By selecting images as the message carrier, this study aims to improve the performance of steganographic methods in terms of concealment, security, capacity, and robustness, thereby achieving efficient and secure secret information transmission while clearly distinguishing image steganography from watermarking techniques.

In practical applications, container images are often subjected to various types of distortions due to factors such as storage space optimization (lossy compression) (*Duan et al., 2023*), which can degrade the quality of the container image and consequently affect the accurate extraction of the embedded secret information. Robustness refers to the ability of the steganographic system to preserve the secret information intact even under such distortions, specifically measuring the similarity between the original secret information and the extracted information after the container image has been compromised.

However, the latest image steganography techniques generally overlook the potential impact of attacks on the container image, resulting in the inability to effectively extract hidden secret information when the container image is compromised. To address this issue, *Xu et al. (2022)* proposed a robust reversible image steganography method that incorporates considerations of image distortion, significantly enhancing the robustness of steganography and enabling successful extraction of hidden information even when the

container image is damaged. *Yang et al. (2024)* introduced a practical robust invertible network for image steganography, improving the model's robustness through the addition of enhancement modules.

In this article, we propose a denoising-enhanced robust invertible neural network termed DERIS. By introducing two identical denoising enhancement modules and employing a denoising-enhanced training strategy, we effectively improve the model's robustness. The main contributions of this work are as follows:

1. We propose a denoising-enhanced robust image steganography framework (DERIS) based on invertible neural networks, which performs robustly under various attacks. This framework significantly enhances the model's robustness and enables it to withstand a wide range of common image distortions.

2. We introduce a denoising-enhanced training strategy that progressively trains both the invertible modules and the denoising enhancement modules, ensuring that the model achieves both reversibility and robustness simultaneously.

3. We design a denoising enhancement module that is integrated before the discrete wavelet transform (DWT) and after the inverse discrete wavelet transform (IDWT) in the backward extraction process. This module enhances robustness during secret image extraction, effectively resists various types of attacks, and improves the quality of the extracted images.

## RELATED WORK

### Traditional image steganography

Traditional image steganography primarily achieves the embedding and transmission of secret information by directly modifying the pixels or statistical properties of an image. These methods are simple to implement, computationally efficient, and capable of achieving a certain level of concealment and data integrity. However, they suffer from limitations in terms of capacity (*Hsu & Wu, 1999*), robustness, image quality, and imperceptibility. Depending on the method of embedding steganographic information, traditional image steganography can generally be classified into two main categories: spatial domain-based methods and transform domain-based methods. In the spatial domain, Least Significant Bit (LSB) replacement embeds secret information into the least significant bits of image pixels. It is easy to implement, offers high embedding capacity, and results in visually imperceptible changes after embedding. However, it is highly sensitive to operations such as compression and filtering, making it vulnerable to detection under common attacks. Pixel Value Differencing (PVD) (*Pan, Li & Yang, 2011*) utilizes the grayscale differences between adjacent pixels for embedding, where areas with larger differences can embed more information while smoother regions carry less. This method improves concealment; however, modifications in flat regions may introduce visible distortions. Histogram Shifting Steganography (*Tsai, Hu & Yeh, 2009*; *Imaizumi & Ozawa, 2014*) embeds secret data by adjusting the grayscale histogram of an image. The information is first converted into a binary bitstream, and appropriate pixel value ranges

are selected for mapping. This approach minimizes distortion to image quality by using small pixel value shifts; however, its concealment and embedding capacity are relatively limited. The Multiple Bit Plane method (*Nguyen, Yoon & Lee, 2006*) embeds information across multiple bit planes by utilizing different color components of an image, such as the red, green, and blue channels. By distributing the hidden data across several bit planes, this method achieves higher hiding capacity and better concealment with minimal impact on image quality. However, increasing the number of color components may lead to a decline in visual fidelity. On the other hand, transform domain-based methods include Discrete Cosine Transform (DCT) steganography (*Khamrui & Mandal, 2013*), where the image is divided into blocks and transformed using DCT, with secret information embedded into middle- or high-frequency coefficients. This method demonstrates good robustness against JPEG compression, although it has limited embedding capacity and involves complex computations. Discrete Wavelet Transform (DWT) (*Ghasemi, Shanbehzadeh & Fassihi, 2011*) decomposes the image into low- and high-frequency sub-bands, with information embedded mainly in the high-frequency components. It provides moderate robustness against various signal processing operations, such as scaling and rotation, though it comes at the cost of high computational complexity. Finally, Discrete Fourier Transform (DFT) steganography (*Chen, 2008*) embeds information by modifying the amplitude or phase components in the frequency domain. It shows strong resistance to geometric attacks such as rotation and cropping, although it may slightly affect visual quality.

## Deep learning-based image steganography

Traditional image steganography employs various spatial domain and transform domain methods to provide multiple approaches for the covert transmission of secret information. Spatial domain-based methods are typically simple to implement and offer high embedding capacity; however, they suffer from limitations in terms of robustness and concealment. In contrast, transform domain-based methods enhance the robustness of embedded information and perform particularly well against compression and geometric attacks. Nevertheless, these methods often involve higher computational complexity and tend to have lower embedding capacity compared to spatial domain approaches. In recent years, with the rapid advancement of deep learning, researchers have proposed deep learning-based image steganography techniques that outperform traditional methods in terms of steganographic quality, robustness, flexibility, and scalability. *Zhu et al. (2018)* introduced HiDDeN, an end-to-end trainable framework that jointly trains an encoder and decoder, demonstrating a degree of robustness against common perturbations such as Gaussian blur and JPEG compression. However, its performance under such conditions is not satisfactory. *Weng et al. (2019)* proposed an innovative video steganography method that leverages a high-capacity convolutional network coupled with temporal residual modeling to markedly increase both the embedding capacity and the imperceptibility of hidden information. Nevertheless, the approach still faces challenges in computational complexity and robustness; its stability, in particular, requires further improvement when confronted with common video-processing operations. *Ahmadi et al. (2020)* achieved

improved watermark embedding and extraction through a residual diffusion mechanism and deep networks, significantly enhancing both invisibility and robustness. Nevertheless, the method still exhibits vulnerability under extreme compression or geometric distortions such as cropping and scaling. *Tancik, Mildenhall & Ng (2020)* proposed the StegaStamp framework, which embeds hyperlinks into physical photographs and successfully extracts them through decoding, innovatively bridging steganography with the physical world and enhancing its practical significance. Despite this innovation, the model lacks resilience against severe lighting changes or camera noise during physical image capture. The same authors also developed a steganography method based on generative adversarial networks (GANs), optimizing the perceptual quality of steganographic images through adversarial training, resulting in highly concealed images that effectively evade detection by analytical tools. However, GAN-based models remain fragile when exposed to strong JPEG compression or filtering operations. *Luo et al. (2020)* improved robustness against unknown distortions by replacing traditional fixed distortion sets with a generator, enabling the model to better handle unpredictable image transformations. While this approach offers greater adaptability, it still struggles to maintain stable performance under combinations of multiple attacks. *Zhang et al. (2023)* proposed an innovative image steganography approach that significantly improves the security and imperceptibility of embedded data by employing a joint adjustment mechanism and multi-task learning framework. Nevertheless, the method entails considerable computational complexity, and its robustness—particularly its resilience to common image processing operations—still requires further enhancement. *Yao et al. (2024)* proposed an end-to-end approach combining GANs and SWT to improve the quality of generated steganographic images. *Hu et al. (2024)* proposed a steganography method based on Stable Diffusion, utilizing zero-shot generation techniques to achieve text-driven steganographic image generation, demonstrating excellent performance in robustness and security. Finally, *Su, Ni & Sun (2024)* developed Steganography StyleGAN, which effectively enhances security, capacity, and visual quality. However, the robustness of the hidden information may be compromised when subjected to common image-processing operations.

## Invertible neural networks

Invertible neural networks are a special class of neural networks characterized by their ability to fully recover input information through reversible operations during both forward and backward propagation. This means that at each layer or module of the network, the input can be completely reconstructed from the output, avoiding the irreversible information loss commonly found in traditional neural networks. *Dinh, Krueger & Bengio (2014)* proposed NICE, the first generative model based on reversible networks, which aimed to perform data density estimation through invertible mappings while supporting efficient generation. NICE utilized coupling layers to bypass the complex Jacobian matrix computations required in conventional density estimation models. However, it exhibits significant shortcomings in terms of robustness, particularly when the input data is subjected to common image degradations such as noise interference or

compression, leading to a noticeable decline in both the quality of the generated results and the accuracy of the extracted hidden information. Subsequent improvements introduced more flexible transformation designs, enhancing model performance while preserving reversibility and computational efficiency. In recent years, invertible neural networks have been widely applied across various domains in deep learning, with representative applications including image generation and super-resolution reconstruction (*Liu et al., 2023*), data hiding and privacy protection (*Yang et al., 2023*), anomaly detection (*Sun et al., 2024*), and Bayesian inference (*Wu, Huang & Zhao, 2023*). Within the field of image steganography, invertible neural networks have demonstrated significant potential. *Jing et al. (2021)* proposed Hi-Net, a model that integrates invertible neural networks with frequency-domain steganography. This method applies wavelet transforms to images, embedding secret information into frequency regions more suitable for concealment. It also leverages a hierarchical network structure combined with generative adversarial network (GAN) techniques (*Goodfellow et al., 2014*) to improve the quality and imperceptibility of steganographic images, offering new perspectives for the development of steganography. However, its robustness is notably insufficient, especially when subjected to common image degradations such as JPEG compression or Gaussian noise, leading to a significant decline in the ability to recover the hidden information. *Lu et al. (2021)* introduced ISN, which utilizes invertible neural networks for large-capacity image steganography. ISN can hide single or multiple images in the pixel domain and fully recover the hidden information through reverse operations, while maintaining high image quality and security. Nevertheless, the method still exhibits vulnerability under practical attacks such as cropping, scaling, and lossy compression, significantly degrading the recovery performance of the embedded data. *Yang et al. (2024)* proposed the PRIS model, a highly robust steganographic framework based on invertible neural networks, particularly effective against various perturbations such as compression and noise. Traditional image steganography faces several limitations in terms of robustness, including vulnerability to image attacks, the trade-off between capacity and robustness, sensitivity to adversarial manipulation, limited recovery capabilities, and the irreversibility of many existing methods. Although deep learning has enabled significant progress in improving robustness and image recovery, challenges such as image distortion, compression, and other types of attacks remain. *Yang, Xu & Liu (2025)* proposed DKiS, an invertible neural network-based image steganography method. It aims to enhance the security and concealment of embedded information by introducing a private key mechanism and a design featuring decay weights. This method not only allows for the recovery of the original image after embedding information but also ensures that the embedded information is difficult for unauthorized third parties to detect or extract. However, it has relatively high computational resource requirements and its robustness still needs improvement. Therefore, enhancing the robustness of image steganography—especially under diverse attack scenarios—remains an active area of research. To address these challenges, we propose DERIS, a denoising-enhanced robust invertible steganographic model that improves robustness, resists various types of attacks, and enhances the quality of extracted images. By introducing identical denoising enhancement modules before the DWT and

after the inverse discrete wavelet transform (IDWT) in the backward extraction process, along with a dedicated denoising-enhanced training strategy, our model achieves state-of-the-art performance in robust image steganography.

### Image denoising techniques

Image denoising plays a crucial role in image steganography, as it directly affects the quality of both information embedding and extraction processes. The primary goal of image steganography is to embed secret information into a host image while maintaining visual indistinguishability and ensuring accurate retrieval of the hidden data. However, in practical applications, the host image may be subjected to various distortions or attacks, such as compression and noise interference, which can hinder the accurate recovery of the embedded information. Therefore, effective image denoising techniques not only improve the quality of the original host image but also enhance the robustness and security of the overall steganographic system. Recently, with the rapid development of deep learning techniques, numerous efficient image denoising methods have emerged, providing powerful tools to improve the reliability and performance of image steganography in complex real-world scenarios. For instance, the deep convolutional neural network-based denoising model DnCNN (*Zhang, Zuo & Zhang, 2021*) performs image denoising by learning a direct mapping from noisy to clean images. The residual learning framework, exemplified by RIDNet (*Anwar, Li & Barnes, 2022*), enhances training efficiency by focusing on learning the noise component rather than the entire image, thereby significantly improving denoising performance. The reinforced residual encoder–decoder network further improves upon traditional encoder–decoder architectures through deeper feature encoding and balanced skip connections (*Liu, Wang & Chen, 2022a*), making it particularly suitable for image denoising in complex environments. Additionally, GAN-based denoising methods, such as Restormer (*Zamir et al., 2022*), utilize adversarial training to enable the generator to learn how to produce noise-free images, while the discriminator works to distinguish between real and generated clean images. Inspired by these recent advancements in image denoising, we propose DERIS, which aims to significantly enhance the model's resistance to image distortions and ensure accurate extraction of hidden information even under various attacks. To provide a clearer overview of the evolution of steganography, this article presents a comparative summary of several key methods in Table 1, highlighting their strengths as well as their remaining limitations.

## METHOD

In this section, we present a detailed description of the network architecture of DERIS (see Fig. 1). Built upon invertible neural networks, the proposed model enhances robustness and improves the quality of extracted images by incorporating two denoising enhancement modules and a dedicated denoising-enhanced training strategy. Notably, when the training images are not subjected to common attacks such as JPEG compression or additive Gaussian noise, the denoising enhancement modules are not activated. In such cases, only the first stage of the denoising-enhanced training strategy is performed. This design choice prevents over-processing of clean images, which could otherwise degrade the performance

**Table 1 Characteristics and limitations of key steganography methods.**

| Reference | Characteristics | Limitations |
|---|---|---|
| NICE (*Dinh, Krueger & Bengio, 2014*) | Efficient generation and density estimation | Lacks robustness to noise and compression |
| Hinet (*Jing et al., 2021*) | Improves stego image quality and imperceptibility | Poor robustness to compression and Gaussian noise |
| ISN (*Lu et al., 2021*) | Large embedding capacity and perfect image recovery | Vulnerable to cropping, scaling, and lossy compression |
| PRIS (*Yang et al., 2024*) | Excellent resistance to compression and noise | Performance fluctuates under compound attacks |
| DKIS (*Yang, Xu & Liu, 2025*) | Strong security and high concealment | High computational cost and limited robustness |
| *Weng et al. (2019)* | High embedding capacity for video; exploits temporal redundancy | Limited robustness to video compression and frame loss |
| Joint (*Zhang et al., 2023*) | Enhanced security and multi-task optimization | High computational complexity; limited robustness to common image processing operations |

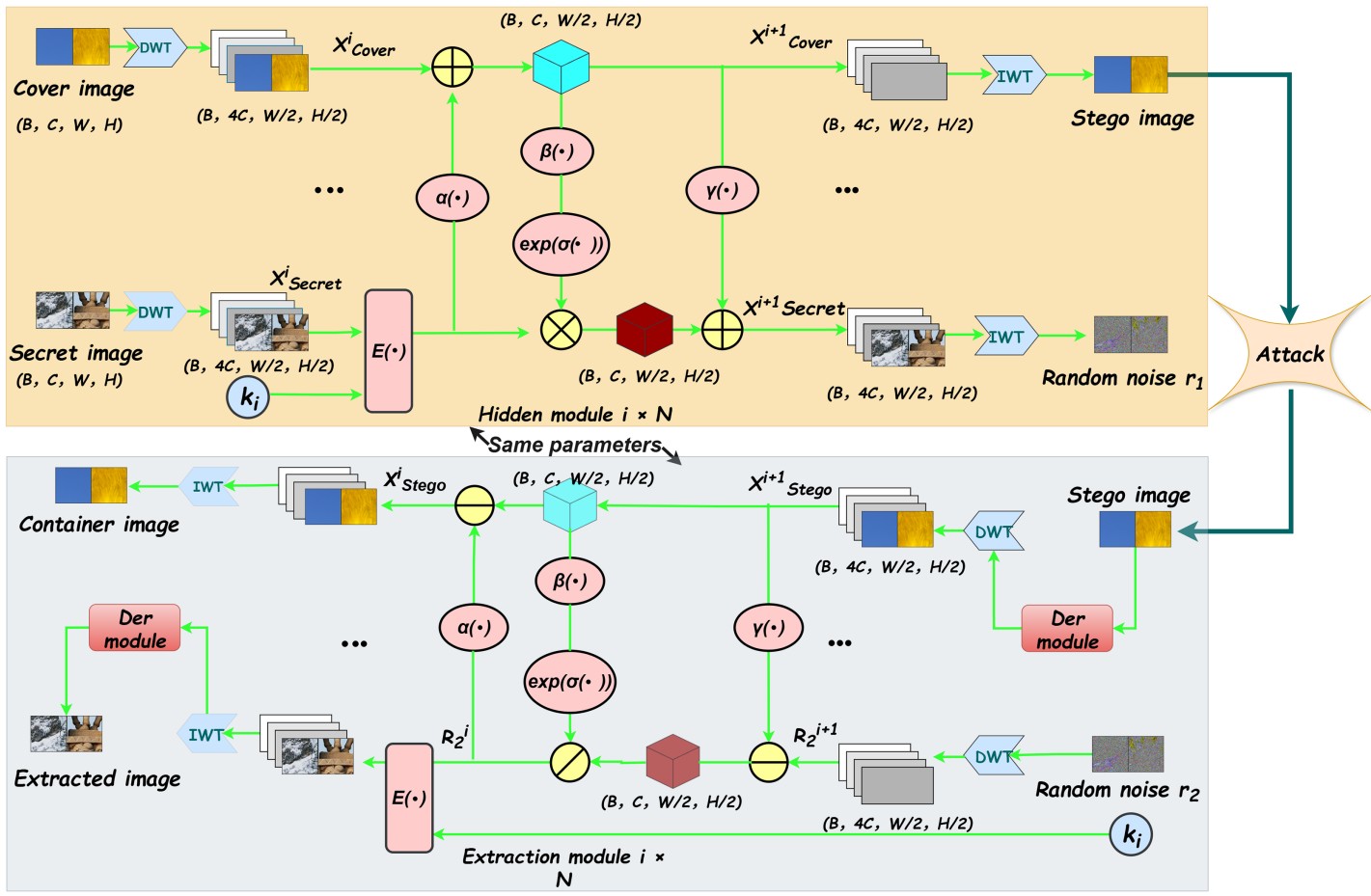

**Figure 1 The framework structure of DERIS.The proposed architecture primarily comprises two types of modules: reversible blocks and denoising enhancement blocks.** The yellow components in the upper section of the figure illustrate the forward hiding process, whereas the gray components in the lower section correspond to the backward extraction process.Each reversible block integrates multiple hiding and extraction modules that share identical network architectures. DWT stands for Discrete Wavelet Transform, which is used to decompose the input image into different frequency sub-bands, while IWT denotes the Inverse Wavelet Transform, employed to reconstruct the image from these sub-bands.

---

**Algorithm 1** Algorithm for the forward hiding process.

Require: $x_{cover}, x_{secret}, k_i$
Ensure: $x_{stego}, r_1$
1: $(x_{cover}) \xrightarrow{\text{DWT}} (x^i_{cover})$,   $(x_{secret}) \xrightarrow{\text{DWT}} (x^i_{secret})$
2: $x^i_{secret}, k_i \xrightarrow{E} (x^i_k)$
3: **for** $i = 1$ to $n$ **do**
4:   $x^i_{cover} \xrightarrow{\text{InvBlock}^i} (x^{i+1}_{cover})$
5:   $x^i_k \xrightarrow{\text{InvBlock}^i} (x^{i+1}_k)$
6: **end for**
7: $(x^{i+1}_{cover}) \xrightarrow{\text{IWT}} (x_{stego})$
8: $(x^{i+1}_k) \xrightarrow{\text{IWT}} (r_1)$
9: **return** $x_{stego}, r_1$

---

**Algorithm 2** Algorithm for the backward extraction process.

**Require:** $x_{stego}, r_2, k_i$
**Ensure:** $x_{container}, x_{extracted}$
1: $(x_{stego}) \xrightarrow{\text{DWT}} (x^i_{stego})$,   $(r^{i+1}_2) \xrightarrow{\text{DWT}} (r^i_2)$
2: **for** $i = n$ downto 1 **do**
3:   $x^{i+1}_{stego} \xrightarrow{\text{InvBlock}^i} (x^i_{stego})$
4:   $r^{i+1}_2 \xrightarrow{\text{InvBlock}^i} (r^i_2)$
5:   $r^{i+1}_2, k_i \xrightarrow{E} (r^i_{k2})$
6: **end for**
7: $x^i_{stego} \xrightarrow{\text{IWT}} (x_{container})$
8: $r^i_2 \xrightarrow{\text{IWT}} (x_{extracted})$
9: **return** $x_{container}, x_{extracted}$

---

of the extraction process. The model is primarily trained on images from the DIV2K (https://data.vision.ee.ethz.ch/cvl/DIV2K/), COCO (https://cocodataset.org/#download), and ImageNet (https://www.image-net.org/challenges/LSVRC/index.php) datasets.

## Overview

Let $x_{cov}, x_s, x_{con}, x_e$, and $k$ denote the cover image, secret image, container image, extracted image, and private key, respectively. The overall architecture of the proposed model is illustrated in Fig. 1. In the forward hiding process, the cover image $x_{cov}$ and the secret image $x_s$, embedded with the private key $k$ through the encoding function $E(\cdot)$, are taken as inputs. These images are first decomposed into low-frequency wavelet sub-bands *via* DWT. Subsequently, the resulting sub-bands are passed through $N$ cascaded hiding blocks. The output of the final hiding block is then processed by IWT to generate the container image $x_{con}$ and the random noise term $r_1$. To provide a clear and structured description of the forward hiding process, the corresponding algorithm is summarized in Algorithm 1. Similarly, in the backward extraction process, the container image $x_{con}$ and the auxiliary random noise $r_2$ are first transformed using DWT, followed by a series of extraction blocks, which reconstruct the extracted image $x_e$. This procedure is detailed in Algorithm 2.

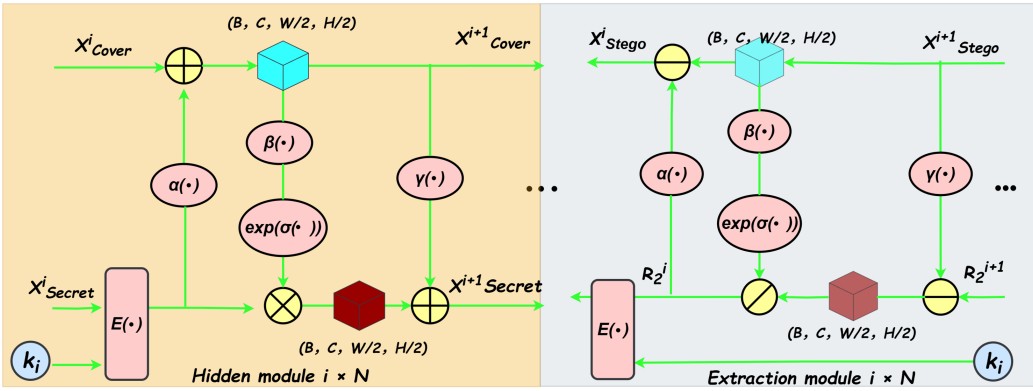

**Figure 2 Illustrates the framework structure of the reversible module.** The hiding module and the extraction module share an identical network architecture. The yellow component on the left depicts the forward hiding process, while the grey component on the right corresponds to the backward extraction process. Within this architecture, $\alpha(\cdot)$, $\beta(\cdot)$, and $\gamma(\cdot)$ represent three convolutional subnetworks with identical structures. $E(\cdot)$ denotes the encoding function, and $k$ represents the private key. The symbol $\sigma$ indicates the sigmoid activation function. Furthermore, $\otimes$, $\oplus$, $\ominus$, and $\oslash$ denote element-wise multiplication, addition, subtraction, and division, respectively.

### Random noise $r_1$ and $r_2$

The random noise $r_1$ is one of the outputs generated during the forward hiding process, while the random noise $r_2$ serves as an input in the backward extraction process and can also be regarded as an auxiliary variable. Suppose the objective is to embed a secret image into a container image while ensuring that the resulting stego image remains visually indistinguishable from the original one. In such a scenario, part of the original information in the container image will inevitably be modified due to the limited embedding capacity of the image. The displaced information corresponds to what we define as the random noise $r_1$. To model this uncertainty in a general and data-independent manner, $r_1$ is sampled from a standard Gaussian distribution, as formally defined in Eq. (1).

$$r_1 \sim \mathcal{N}(0, 1). \tag{1}$$

In the experiments conducted in this work, we assume that $\mu_0 = 0$ and $\sigma_0^2 = 1$. This prior distribution is designed to be consistent with the distribution of the random noise $z_2$, thereby compensating for the information lost during the reconstruction process.

During the backward extraction process, the container image $x_{\mathrm{con}}$ serves as the sole available input. If the extraction of the image $x_e$ relies solely on $x_{\mathrm{con}}$, the unknown nature of the random noise $z_1$ renders the problem ill-posed, in the sense that infinitely many possible solutions for $x_e$ could be generated from the same $x_{\mathrm{con}}$. To address this issue, we introduce an additional random noise term $r_2$, assumed to follow the same distribution as $r_1$, thereby regularizing the solution space and ensuring a well-defined extraction process.

### Invertible block

The forward hiding module and the backward extraction module within the reversible block share an identical network architecture, as illustrated in Fig. 2. This structure

processes two input images and generates two output images during both the forward and reverse operations, as depicted in Fig. 2. The computational formulas for the forward hiding process are detailed below (see Eqs. (2), (3), (4), (5)). The inputs to this process include the cover image $x_{cov}^i$ and the secret image $x_s^i$. Initially, the secret image $x_s^i$, embedded with a private key $k$ through the encoding function $E(\cdot)$, is processed through the function $\alpha(\cdot)$. The resulting image is then added element-wise to the cover image $x_{cov}^i$ to produce $x_{cov}^{i+1}$. Subsequently, $x_{cov}^{i+1}$ undergoes sequential processing by the function $\beta(\cdot)$ followed by an activation function $\sigma$, and the output is scaled using the exponential function exp to generate an intermediate result $x_{scaled}$. Finally, $x_{scaled}$ is multiplied element-wise with the secret image $x_s^i$, and the resulting product is added element-wise to $\gamma(x_{cov}^{i+1})$, yielding the updated secret image $x_s^{i+1}$.

$$X_k^i = E(X_s^i, k_i), \tag{2}$$
$$X_{cov}^{i+1} = X_{cov}^i \oplus \alpha(X_s^i), \tag{3}$$
$$X^{scaled} = \exp(\sigma(\beta(X_{cov}^{i+1}))), \tag{4}$$
$$X_s^{i+1} = (X_s^i \otimes X^{scaled}) \oplus \gamma(X_{cov}^{i+1}), \tag{5}$$

The computational formulas for the backward extraction process are detailed as follows (see Eqs. (6), (7), (8), (9), (10)). The inputs to this process include a random noise $z^{i+1}$ and the container image $x_{con}^{i+1}$. Initially, the function $\gamma(\cdot)$ is applied to $x_{con}^{i+1}$, and the result is subtracted element-wise from $z^{i+1}$ to obtain $r_{x_i}$. Subsequently, $x_{con}^{i+1}$ undergoes sequential processing through the function $\beta(\cdot)$ followed by an activation function $\sigma$, and the output is scaled using the exponential function exp to produce an intermediate variable $x_{scaled}$. Afterwards, $r_{x_i}$ is divided element-wise by $x_{scaled}$ to derive $r_i$. Applying the function $\alpha(\cdot)$ to $r_i$, its output is then subtracted element-wise from $x_{con}^{i+1}$ to reconstruct the preceding container image $x_{con}^i$. Finally, the encrypted extracted image $x_k^i$ is processed through the decoding function $E^{-1}(\cdot)$, resulting in the decrypted extracted image $x_s^i$.

$$r_x^i = \gamma(x_{con}^{i+1}) \ominus z^{i+1}. \tag{6}$$
$$x_{scaled} = \exp(\sigma(\beta(x_{con}^{i+1}))). \tag{7}$$
$$r^i = r_x^i \oslash x_{scaled}. \tag{8}$$
$$x_{con}^i = x_{con}^{i+1} \ominus \alpha(r^i). \tag{9}$$
$$x_s^i = E^{-1}(x_K^i, K). \tag{10}$$

## Loss function

Our research primarily focuses on two distinct types of perceptual similarity. The first type involves the difference between the cover image $x_{cov}$ and the container image $x_{con}$, which we denote as C-pairs. The second type pertains to the difference between the secret image $x_s$ and the extracted image $x_e$, referred to as S-pairs.

To quantify these similarities, we employ the Peak Signal-to-Noise Ratio (PSNR) metric. Higher PSNR values indicate greater similarity between the compared images. Specifically, $PSNR_C$ and $PSNR_S$ denote the PSNR values computed for C-pairs and

S-pairs, respectively. Our objective is to maximize both $\text{PSNR}_C$ and $\text{PSNR}_S$, thereby ensuring high fidelity in both cover-to-container and secret-to-extracted image transformations.

To quantitatively assess the discrepancies between these pairs, we introduce two loss functions, denoted as $L_c$ and $L_s$. The precise formulations are presented below:

$$L_c = \sum_p (x_c^{(p)} - x_h^{(p)})^2. \tag{11}$$

$$L_s = \sum_p (x_s^{(p)} - x_e^{(p)})^2. \tag{12}$$

Among them, $x_p$ represents the value of image $x$ at pixel position $p$. The loss functions $L_c$ and $L_s$ are used to evaluate the similarity of C-pairs and S-pairs, respectively. The final loss function is the weighted sum of these two losses:

$$L = \lambda_c L_c + \lambda_s L_s \tag{13}$$

where $\lambda_c$ and $\lambda_s$ are the weighting coefficients for the two loss terms.

## Denoising enhancement training strategy

The purpose of the denoising enhancement training strategy is to strike a balance between the inherent reversibility of the network and the partial irreversibility introduced by the denoising enhancement (Der) module, thereby fulfilling the dual requirements of reversibility and robustness in image steganography. The core idea of this strategy is to progressively train both the reversible block and the Der module to ensure a harmonious integration of information fidelity and noise resilience. A detailed description of each training phase is as follows:

**Step 1: Pre-training the reversible block.**

In this initial phase, only the reversible block is activated and involved in both forward and backward propagation, while the parameters of the Der module are completely ignored. By training the reversible block in isolation, the network achieves strict reversibility at this stage—ensuring that the input images can be exactly reconstructed after being processed through the network. This step establishes a lossless information-preserving foundation for the overall architecture.

**Step 2: Pre-training the Der module.**

In this phase, the Der module is gradually integrated into the training pipeline without compromising the reversibility already achieved. All components of the proposed DERIS framework are enabled during forward propagation; however, the parameters of the reversible block are frozen and remain fixed during backpropagation. Only the parameters of the Der module are updated. This allows the network to begin benefiting from the performance enhancement provided by the Der module, while preserving the stability of the previously learned reversible mapping. As a result, the network begins to exhibit a controlled degree of irreversibility, while maintaining the integrity of the reversible foundation.

**Step 3: Joint fine-tuning of the reversible block and the Der module.**

In the final phase, both the reversible block and the Der module are fully activated, and all network parameters are jointly optimized through backpropagation. This fine-tuning stage enables the different components of the network to adapt to each other and work in a more synergistic manner. Through this collaborative optimization, the network not only satisfies the fundamental requirements of image steganography—namely, reversibility and information fidelity—but also acquires enhanced robustness against various types of distortions and attacks. This results in an optimal trade-off between reversibility and irreversibility, achieving both high-quality reconstruction and strong resistance to real-world perturbations.

## Der module

During the backward extraction process, we incorporate denoising enhancement modules both before DWT and after IDWT, as illustrated in Fig. 1. These two modules share an identical network architecture, which not only reduces the overall model complexity but also alleviates the strict reversibility constraints of the network structure, thereby enhancing the robustness of the system. The denoising enhancement module is built upon a residual dense block, which follows a densely connected stacked structure. This design is inspired by both DenseNet and Residual Network (ResNet) architectures, integrating the advantages of dense connectivity and residual learning. Specifically, the input to each layer consists of the outputs from all preceding layers, in addition to the feature maps generated by the current layer. This mechanism enables direct information flow across layers, enhancing feature reuse and improving the representational capacity of the network. As a result, the proposed architecture achieves superior performance in terms of both image quality restoration and noise suppression, while also facilitating more efficient training.

The denoising enhancement module is placed before DWT with the aim of reducing perturbations present in the container image, thereby providing a cleaner and more reliable input for the subsequent extraction network. By enhancing the quality of the distorted container image, this module improves the overall extraction performance and mitigates the adverse effects of noise on the extracted results. As a result, the pre-enhancement module enhances the robustness of the entire extraction pipeline, facilitating more accurate and stable reconstruction of the original secret image, even under challenging distortion conditions.

The denoising enhancement module is also placed after the IDWT with the aim of improving the quality of the extracted image, making it closer to the original secret image. This module enhances the visual fidelity of the extracted image by refining its structural details and increasing its clarity, thereby effectively mitigating the adverse effects of noise introduced during the extraction process. Through this enhancement mechanism, the post-processing module not only restores the perceptual quality of the extracted image but also strengthens the robustness of the overall extraction procedure. As a result, the final reconstructed secret image is more accurate, reliable, and visually realistic, demonstrating improved performance under various distortion conditions.
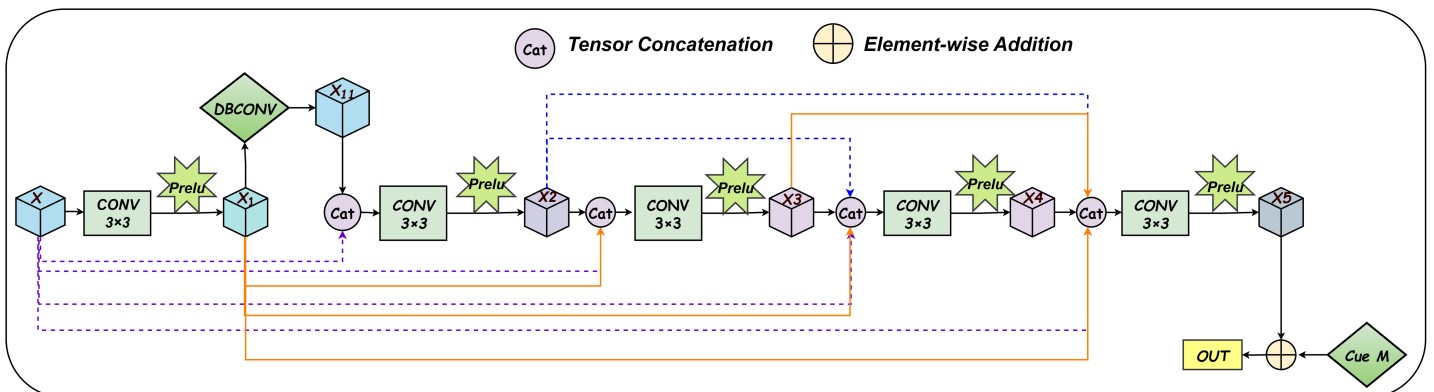

**Figure 3 Detailed structure diagram of the denoising enhancement module.**

The denoising enhancement module integrates a Depthwise-Mobile Inverted Bottleneck Convolution (DBConv) module into the first layer of the fixed, densely connected stacked architecture, as illustrated in Fig. 3. The overall network structure is further detailed in Fig. 4. The DBConv module follows an expand-depthwise-contract paradigm: it first expands the number of feature channels, processes them using depthwise convolution, and then compresses the channels back to the original dimension. This architectural design enables the network to better focus on salient features, particularly when extracting meaningful information from noisy data. The DBConv module employs depthwise separable convolutions, where each input channel is processed independently through a depthwise convolution, followed by a pointwise convolution that combines the output across channels. This approach significantly reduces both computational complexity and the total number of model parameters. For image enhancement and denoising tasks, the depthwise convolution allows the network to concentrate on fine-grained local features, thereby improving the precision of feature representation. This capability facilitates effective noise suppression while preserving and even enhancing important image details.

The Squeeze-and-Excitation (SE) module enhances the discriminative capability of the network by adaptively recalibrating channel-wise feature responses. In the context of image enhancement, the SE module enables the network to emphasize more informative channels, thereby effectively preserving key structural and semantic features. This mechanism is particularly beneficial for maintaining critical image content while suppressing noise, ultimately contributing to improved overall image quality. Residual connections facilitate stable gradient propagation through the network, mitigating the vanishing gradient problem. They help retain the structural integrity of the input image while allowing the model to progressively refine the noise suppression effect across layers. Additionally, the Dropout layer improves the model's generalization performance by preventing overfitting during training. By randomly deactivating a subset of neurons, Dropout encourages the network to learn more robust and distributed feature representations. This ensures consistent and reliable performance when handling images with varying levels of noise and degradation.

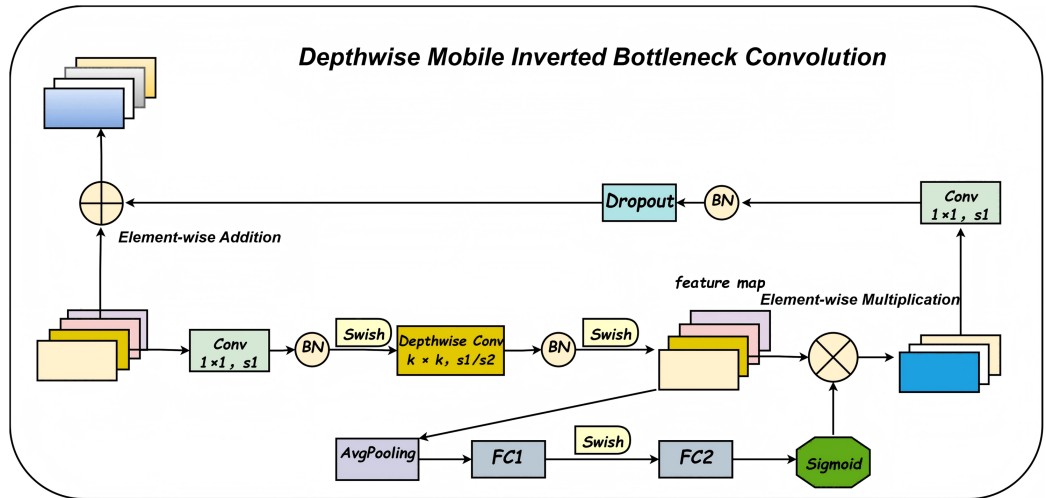

**Figure 4 Convolutional structure diagram of deep movement inversion bottleneck.**

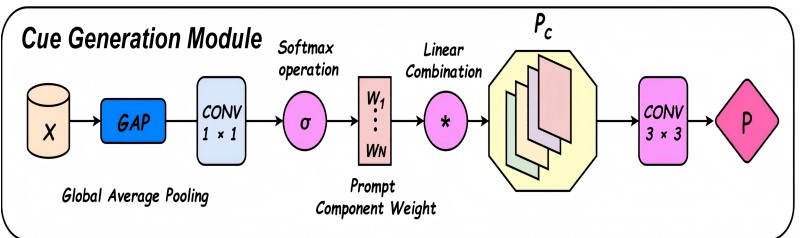

**Figure 5 Prompt to generate the module structure diagram.**

The denoising enhancement module incorporates a Cue Generation Module (Cue M) at the end of a fixed, densely connected stacked architecture, as illustrated in Fig. 3. The overall network structure is further detailed in Fig. 5. The Cue M module generates cue features based on the global characteristics of the input image, dynamically assigning different weights to specific feature channels. This enables the network to focus more selectively on salient features during the processing of diverse input images. Consequently, the proposed model demonstrates greater adaptability to carrier images affected by varying levels of distortion or noise. This adaptability enhances the consistency of the extracted features and improves the overall robustness of the system. Furthermore, the Cue M module effectively suppresses noise and provides informative cues to guide the decoding and reconstruction process. These cues assist the network in more accurately recovering the secret image during the extraction stage. By mitigating the adverse effects of perturbations in the container image, the proposed design significantly enhances the model's resistance to interference and noise. As a result, the extracted secret images are notably closer to the original, demonstrating superior reconstruction performance under challenging conditions.

**Mathematical description of the Der module** Let the input feature map be denoted as $x \in \mathbb{R}^{C \times H \times W}$. The Der module enhances this input through the following operations, where:

- $\sigma$ denotes the Leaky ReLU activation function,
- $\phi$ denotes the Swish activation function,
- $\rho$ denotes the Sigmoid activation function,
- GAP denotes global average pooling,
- $FC_1$ and $FC_2$ denote fully connected layers,
- $W_l$ denotes the linear transformation matrix.

1. **Initial convolution:**

$$x_1 = \sigma\left(\text{Conv}_{3\times3}^{(1)}(x)\right). \tag{14}$$

2. **Depthwise Mobile Inverted Bottleneck Convolution module (DBConv):** This block applies expand layer, depthwise convolution layer, Squeeze-and-Excitation attention, and project layer:

$$x_e = \phi\left(\text{BN}\left(\text{Conv}_{1\times1}(x_1)\right)\right). \tag{15}$$

$$x_{dw} = \phi\left(\text{BN}\left(\text{DWConv}_{3\times3}(x_e)\right)\right). \tag{16}$$

$$s = \rho\left(FC_2 \cdot \phi\left(FC_1 \cdot \text{GAP}(x_{dw})\right)\right). \tag{17}$$

$$x_{se} = s \cdot x_{dw}. \tag{18}$$

$$x_{proj} = \text{BN}\left(\text{Conv}_{1\times1}(x_{se})\right). \tag{19}$$

$$x_{db} = x_1 + x_{proj} \quad \text{(if input shape matches).} \tag{20}$$

3. **Residual dense feature fusion:** The outputs from previous layers are concatenated and convolved in a dense manner:

$$x_2 = \sigma\left(\text{Conv}_{3\times3}^{(2)}([x, x_{db}])\right). \tag{21}$$

$$x_3 = \sigma\left(\text{Conv}_{3\times3}^{(3)}([x, x_1, x_2])\right). \tag{22}$$

$$x_4 = \sigma\left(\text{Conv}_{3\times3}^{(4)}([x, x_1, x_2, x_3])\right). \tag{23}$$

$$x_5 = \text{Conv}_{3\times3}^{(5)}([x, x_1, x_2, x_3, x_4]). \tag{24}$$

4. **Cue Generation Module (Cue M):** The Cue module is composed of global average pooling (GAP), followed by a linear transformation (or $1 \times 1$ convolution), Softmax-normalized weight generation, weighted combination of learnable prompt vectors, spatial interpolation, and finally a $3 \times 3$ convolution to produce the output:

$$w = \text{Softmax}\left(W_L \cdot \text{GAP}(x)\right). \tag{25}$$

$$P_c = \sum_{l=1}^{L} w_l \cdot P_l. \tag{26}$$

$$p = \text{Interpolate}(P_c, (H, W)). \tag{27}$$

$$p' = \mathrm{Conv}_{3\times3}(p). \tag{28}$$

5. **Output:**

$$y = x_5 + p'. \tag{29}$$

We can draw an analogy between the strict reversibility of reversible neural networks and the transmission of noise, by comparing them to a rigid metal pipe and a flexible hose with buffering capabilities. A rigid metal pipe directly transfers all impurities and disturbances in the water flow to the other end without any attenuation or filtering. In contrast, a flexible hose with buffering capabilities can absorb and attenuate some of these impurities and shocks, resulting in a smoother and cleaner water flow at the receiving end. Introducing a denoising and enhancement module into the network is analogous to incorporating a "flexible hose" rather than a "metal pipe" into the architecture. This design enables the network to absorb and mitigate noise during the information transmission process, rather than propagating it in its entirety to the output. Consequently, the network exhibits greater flexibility and improved robustness in noisy environments.

## Steganographic key

To enhance the security and confidentiality of the embedded data, this study introduces a steganographic key mechanism into the processing pipeline of the secret information. The image content is effectively encrypted through the encoding function $E(\cdot)$, as illustrated in Fig. 1. Specifically, the encoding process consists of two critical stages, formulated as follows: First, the original secret image undergoes a block-based reordering operation using the scrambling key $k_{si}$. The image is partitioned into non-overlapping $4 \times 4$ sub-blocks, which are then rearranged according to a permutation sequence defined by $k_{si}$. This step significantly increases the complexity of the information-hiding process, making it more difficult for unauthorized users to reconstruct or extract the embedded content. Second, the scrambled image is further processed through a pixel-wise element-wise multiplication with the multiplicative key matrix $k_{mi}$. Here, $k_{mi}$ is a matrix of the same dimensions as the image blocks, with each element independently and uniformly taking values from $\{-1, 1\}$. This operation further obfuscates the image content, enhancing its resistance to visual and statistical analysis. This dual-layer encryption mechanism not only improves the security of the transmitted images but also enhances their imperceptibility in steganographic applications, thereby significantly strengthening the system's robustness against various types of potential attacks.

$$X_k^{i+1} = [\mathrm{shuffle}(X_s^i, k_{si})] \otimes k_{mi}. \tag{30}$$

The system employs a key derivation mechanism based on the SHA-256 hash algorithm to generate the required steganographic key parameters. Specifically, users can preset an initial key of arbitrary form (*e.g.*, a string or password), which is first processed by the SHA-256 algorithm to produce a fixed-length 256-bit digest. This step enhances the uniformity and collision resistance of the key material. The resulting 256-bit value is then

used as a seed for a cryptographically secure pseudo-random number generator (PRNG), which generates two essential key components: $k_{si}$ and $k_{mi}$. The parameter $k_{si}$ governs the reordering sequence of the image blocks during scrambling, while $k_{mi}$ determines the pixel-wise multiplication operations applied in the subsequent encryption stage. Together, these components constitute a complete steganographic key system. This hash-based key generation approach offers several advantages, including strong security, reproducibility, and resilience against brute-force attacks. Moreover, it ensures that authorized users can accurately reconstruct the original image, while unauthorized access yields no meaningful information. Overall, the proposed encryption framework seamlessly integrates image processing techniques with cryptographic principles, providing an efficient and secure solution for digital steganography.

# EXPERIMENTS

## Experimental setup

We train our DERIS model using the DIV2K (High Resolution Images) dataset (*Agustsson & Timofte, 2017*), and evaluate it on the DIV2K testing dataset using an RTX 3090, unless otherwise specified. The input images are cropped to a resolution of $224 \times 224$ pixels. Random cropping is applied to the training dataset to improve the model's generalization ability, while center cropping is used for the testing dataset to ensure that the evaluation results are not affected by random factors. Training follows the Denoising Enhancement Training Strategy across three stages, each of 1,800 epochs (5,400 epochs in total). Stage 1: Pre-training the reversible block—only the reversible module is trained while the denoising-enhancement module is disabled. Stage 2: Pre-training the Der module—only the Der module is trained with all reversible-block parameters frozen. Stage 3: Joint fine-tuning of the reversible block and the Der module—both modules are trained simultaneously. The overall runtime is approximately 195 h, averaging 130 s per epoch. We use the Adam optimizer with $\beta_1 = 0.9$ and $\beta_2 = 0.99$. The training batch size is set to 4, and the validation batch size is set to 2. The initial learning rates for the three stages are set to $10^{-4.5}$, $10^{-4.5}$, and $10^{-5.5}$, respectively, and the learning rate is halved every 200 epochs. Both $\lambda_c$ and $\lambda_s$ are set to 1. The full DERIS network comprises 4.48 million parameters, occupying approximately 51.48 MB of storage.

## Evaluation metrics

Six metrics can be used to measure the quality of the cover-image/container-image pairs and the secret-image/recovered-secret-image pairs, including Peak Signal-to-Noise Ratio (PSNR) (*Wang et al., 2019*), Structural Similarity Index (SSIM) (*Nilsson & Akenine-Möller, 2020*), Root Mean Square Error (RMSE) (*Gonzalez & Woods, 2002*), Mean Absolute Error (MAE) (*Gonzalez & Woods, 2002*), Learned Perceptual Image Patch Similarity (LPIPS) (*Zhang et al., 2018*), and Normalized Cross-Correlation (NCC) (*Gonzalez & Woods, 2018*).

## Ablation experiment

In the ablation study, we mainly conducted experiments to evaluate the necessity of the denoising-enhanced training strategy, the effectiveness of the denoising-enhancement

**Table 2 An ablation study is conducted on each model component under Gaussian noise perturbation with $\sigma = 10$ and JPEG compression with QF = 80, including the denoising-enhancement module before the discrete wavelet transform, the denoising-enhancement module after the inverse discrete wavelet transform, and the denoising-enhanced training strategy.**

| Staged | Pre-Der | Post-Der | PSNR-C ↑ | SSIM-C ↑ | APD-C ↓ | RMSE-C ↓ | LPIPS-C ↓ | NCC-C ↑ |
|---|---|---|---|---|---|---|---|---|
| × | × | × | 30.68 | 0.7996 | 6.42 | 8.43 | 0.0441 | 0.9913 |
| ✓ | × | ✓ | 31.96 | 0.8198 | 5.12 | 7.14 | 0.0345 | 0.9927 |
| ✓ | ✓ | × | 31.78 | 0.8154 | 5.35 | 7.35 | 0.0369 | 0.9925 |
| × | ✓ | ✓ | 31.52 | 0.8176 | 5.58 | 7.59 | 0.0375 | 0.9921 |
| ✓ | ✓ | ✓ | 32.43 | 0.8237 | 4.73 | 6.72 | 0.0259 | 0.9931 |

| Staged | Pre-Der | Post-Der | PSNR-S ↑ | SSIM-S ↑ | APD-S ↓ | RMSE-S ↓ | LPIPS-S ↓ | NCC-S ↑ |
|---|---|---|---|---|---|---|---|---|
| × | × | × | 28.88 | 0.7892 | 7.76 | 10.36 | 0.0567 | 0.9901 |
| ✓ | × | ✓ | 29.97 | 0.8097 | 6.54 | 9.15 | 0.0489 | 0.9914 |
| ✓ | ✓ | × | 29.75 | 0.8176 | 6.84 | 9.48 | 0.0501 | 0.9909 |
| × | ✓ | ✓ | 29.53 | 0.8345 | 7.12 | 9.69 | 0.0529 | 0.9911 |
| ✓ | ✓ | ✓ | 30.24 | 0.8347 | 6.16 | 8.76 | 0.0445 | 0.9918 |
| × | × | × | 30.61 | 0.7991 | 7.96 | 8.56 | 0.0567 | 0.9899 |
| ✓ | × | ✓ | 30.97 | 0.7996 | 7.86 | 8.41 | 0.0551 | 0.9916 |
| ✓ | ✓ | × | 30.91 | 0.7993 | 7.89 | 8.36 | 0.0559 | 0.9914 |
| × | ✓ | ✓ | 30.79 | 0.7994 | 7.81 | 8.69 | 0.0571 | 0.9908 |
| ✓ | ✓ | ✓ | 32.24 | 0.8219 | 4.95 | 6.96 | 0.0315 | 0.9928 |
| × | × | × | 28.74 | 0.7884 | 8.11 | 11.21 | 0.0609 | 0.9899 |
| ✓ | × | ✓ | 29.88 | 0.8086 | 6.41 | 9.19 | 0.0497 | 0.9923 |
| ✓ | ✓ | × | 29.61 | 0.8049 | 6.97 | 9.57 | 0.0515 | 0.9911 |
| × | ✓ | ✓ | 29.31 | 0.8067 | 7.05 | 9.38 | 0.0536 | 0.9908 |
| ✓ | ✓ | ✓ | 30.12 | 0.8313 | 6.24 | 8.64 | 0.0457 | 0.9914 |

module in improving robustness, the necessity of incorporating both the DBConv module and Cue M into the denoising-enhancement module, and the optimal choices for $\lambda_1$ and $\lambda_2$. First, we explore the necessity of the denoising-enhanced training strategy and the effectiveness of the denoising-enhancement module in improving robustness. As shown in Table 2, it contains four sub-tables: the first two report experiments under Gaussian noise perturbation with $\sigma = 10$, while the latter two report experiments under JPEG compression with QF = 80. In each table, the second row corresponds to adding only the denoising-enhancement module after the inverse discrete wavelet transform, based on the denoising-enhanced training strategy. The third row corresponds to adding only the denoising-enhancement module before the discrete wavelet transform. The fourth row corresponds to adding both denoising-enhancement modules—before the discrete wavelet transform and after the inverse discrete wavelet transform—without using the denoising-enhanced training strategy. Compared to the baseline in the first row, the PSNR, SSIM, and NCC values are improved, while the APD, RMSE, and LPIPS values are reduced. Among all configurations, the best results are achieved only when the denoising-enhanced training strategy is used together with the denoising-enhancement modules placed both before the discrete wavelet transform and after the inverse discrete

wavelet transform. Under Gaussian noise perturbation with $\sigma = 10$, PSNR-C reaches 32.43 dB and PSNR-S reaches 30.24 dB, while under JPEG compression with QF = 80, PSNR-C reaches 32.24 dB and PSNR-S reaches 30.12 dB. This indicates that placing the denoising-enhancement module before the discrete wavelet transform helps reduce noise in the container image, resulting in higher-quality images entering the extraction process. It enables the reversible network to better resist noise introduced by distortions in the container image, mitigating the propagation of such noise to a certain extent and thereby reducing interference during subsequent extraction. Placing the denoising-enhancement module after the inverse discrete wavelet transform allows for final optimization after image extraction, making the extracted secret image visually closer to the original secret image. It can further eliminate residual fine-grained noise, improving the clarity and similarity of the extracted image. By enhancing the quality of the extracted image, the overall performance of the steganographic system is improved, ensuring that the extracted image is closer to the true secret information.

Second, we explore the necessity of incorporating the DBConv module and the Cue M into the denoising enhancement module. To evaluate their effectiveness, we conducted experiments under Gaussian noise perturbation with $\sigma = 10$ and JPEG compression with QF = 80, where only the DBConv module or only the Cue M module was added. As shown in Table 3, it contains four sub-tables: the first two report experiments under Gaussian noise perturbation with $\sigma = 10$, while the latter two report experiments under JPEG compression with QF = 80. Under Gaussian noise perturbation with $\sigma = 10$, when only the DBConv module was added, PSNR-C increased by 0.75 dB and PSNR-S increased by 0.41 dB compared to the baseline. When only the Cue M module was added, PSNR-C increased by 0.91 dB and PSNR-S increased by 0.86 dB compared to the baseline. The best performance was achieved when both the DBConv module and the Cue M module were included, with PSNR-C reaching 32.43 dB and PSNR-S reaching 30.24 dB. As shown in Fig. 6, the locally magnified images are visually the clearest when both modules are added. Under JPEG compression with QF = 80, when only the DBConv module was added, PSNR-C increased by 0.66 dB and PSNR-S increased by 0.26 dB compared to the baseline. When only the Cue M module was added, PSNR-C increased by 0.91 dB and PSNR-S increased by 1.58 dB. When both the DBConv module and the Cue M module were incorporated, the best performance was achieved, with PSNR-C reaching 32.08 dB and PSNR-S reaching 30.11 dB.

Lastly, we investigate the impact of different weight coefficients, $\lambda_c$ and $\lambda_s$, on the Cover/Container image pair and the Secret/Extracted image pair. Experiments were conducted under Gaussian noise perturbation with $\sigma = 10$ and JPEG compression with QF = 80 to determine the most suitable values for $\lambda_c$ and $\lambda_s$. As shown in Table 4, when $\lambda_c$ is too large, the PSNR of the Secret/Extracted image pair becomes excessively low. Similarly, if $\lambda_s$ is too large, the PSNR of the Cover/Container image pair also decreases significantly. Experimental results demonstrate that the overall performance of the evaluation metrics for both image pairs is optimal only when both $\lambda_c$ and $\lambda_s$ are set to 1.

**Table 3 Under Gaussian noise perturbation with $\sigma = 10$ and JPEG compression with QF = 80, an ablation study is performed to assess the impact of incorporating the DBConv module and the Cue M module.**

| DBConv | Cue M | PSNR-C ↑ | SSIM-C ↑ | APD-C ↓ | RMSE-C ↓ | LPIPS-C ↓ | NCC-C ↑ |
|--------|-------|----------|----------|---------|----------|-----------|---------|
| ✗ | ✗ | 31.21 | 0.7876 | 6.32 | 8.32 | 0.0388 | 0.9911 |
| ✓ | ✗ | 31.96 | 0.7994 | 5.97 | 7.69 | 0.0345 | 0.9918 |
| ✗ | ✓ | 32.12 | 0.8143 | 5.45 | 7.47 | 0.0281 | 0.9925 |
| ✓ | ✓ | 32.43 | 0.8237 | 4.73 | 6.72 | 0.0259 | 0.9931 |

| DBConv | Cue M | PSNR-S ↑ | SSIM-S ↑ | APD-S ↓ | RMSE-S ↓ | LPIPS-S ↓ | NCC-S ↑ |
|--------|-------|----------|----------|---------|----------|-----------|---------|
| ✗ | ✗ | 29.16 | 0.7646 | 7.85 | 10.64 | 0.529 | 0.9908 |
| ✓ | ✗ | 29.57 | 0.7893 | 6.95 | 9.78 | 0.515 | 0.9903 |
| ✗ | ✓ | 30.02 | 0.8043 | 6.98 | 9.56 | 0.449 | 0.9913 |
| ✓ | ✓ | 30.24 | 0.8347 | 6.16 | 8.76 | 0.0445 | 0.9918 |
| ✗ | ✗ | 30.91 | 0.7846 | 7.15 | 8.64 | 0.0411 | 0.9887 |
| ✓ | ✗ | 31.57 | 0.7987 | 6.98 | 8.48 | 0.0385 | 0.9896 |
| ✗ | ✓ | 31.82 | 0.7949 | 6.87 | 8.36 | 0.0317 | 0.9911 |
| ✓ | ✓ | 32.08 | 0.8197 | 6.16 | 8.12 | 0.0287 | 0.9925 |
| ✗ | ✗ | 28.87 | 0.7632 | 7.97 | 10.81 | 0.541 | 0.9898 |
| ✓ | ✗ | 29.13 | 0.7754 | 7.85 | 10.55 | 0.489 | 0.9909 |
| ✗ | ✓ | 29.85 | 0.7918 | 6.98 | 10.16 | 0.424 | 0.9916 |
| ✓ | ✓ | 30.11 | 0.8113 | 6.16 | 9.78 | 0.0945 | 0.9921 |

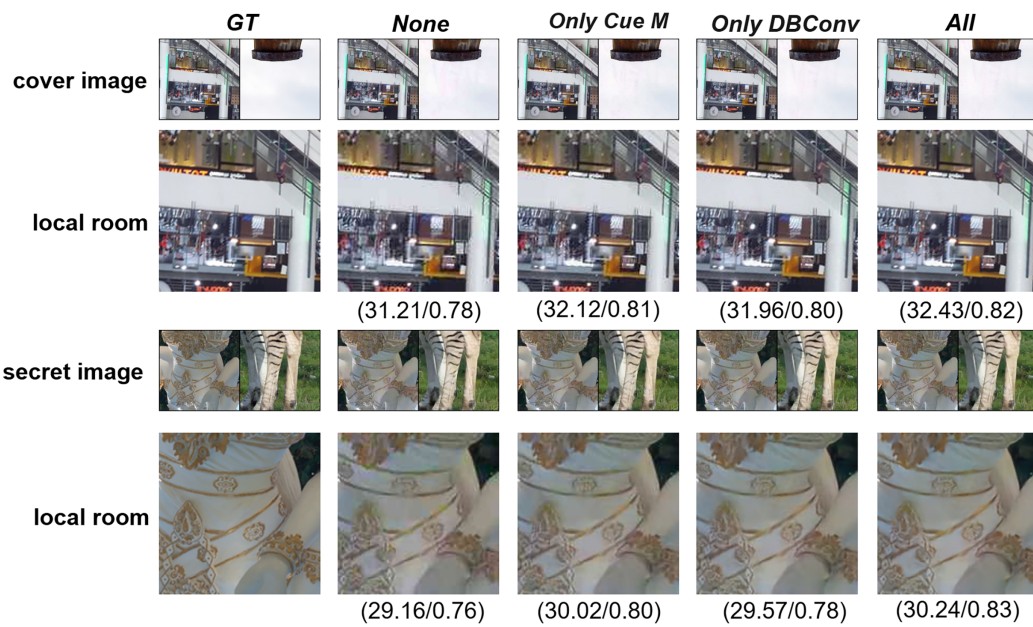

**Figure 6 Ablation study under Gaussian noise perturbation with $\sigma = 10$, evaluating the effects of incorporating the DBConv module and the Cue M module.** Even rows show enlarged local details.

**Table 4 Performance comparison of different $\lambda_c$ and $\lambda_s$ under Gaussian noise perturbation with $\sigma = 10$ and JPEG compression with QF = 80.**

| $\lambda_c$ | 0.2 | 0.4 | 0.8 | 1.0 | 1.2 | 1.4 | 1.8 |
|---|---|---|---|---|---|---|---|
| $\lambda_s$ | 1.8 | 1.6 | 1.2 | 1.0 | 0.8 | 0.6 | 0.2 |
| PSNR-C | 30.68 | 31.41 | 31.91 | **32.43** | 33.57 | 35.68 | 37.76 |
| PSNR-S | 32.13 | 31.56 | 30.97 | **30.24** | 28.89 | 26.96 | 23.67 |
| SSIM-C | 0.9201 | 0.9215 | 0.9221 | **0.9239** | 0.0187 | 0.0098 | 0.0028 |
| SSIM-S | 0.9173 | 0.9187 | 0.9191 | **0.9197** | 0.9241 | 0.9484 | 0.9649 |
| LPIPS-C | 0.0686 | 0.0584 | 0.0475 | **0.0386** | 0.0187 | 0.0098 | 0.0028 |
| LPIPS-S | 0.0489 | 0.0523 | 0.0637 | **0.0714** | 0.0934 | 0.1042 | 0.2151 |
| NCC-C | 0.9915 | 0.9921 | 0.9932 | **0.9944** | 0.9953 | 0.9967 | 0.9988 |
| NCC-S | 0.9935 | 0.9924 | 0.9913 | **0.9907** | 0.9867 | 0.9841 | 0.9701 |
| PSNR-C | 30.58 | 31.31 | 31.85 | **32.38** | 33.17 | 35.56 | 37.46 |
| PSNR-S | 32.27 | 31.49 | 29.83 | **30.08** | 28.98 | 27.96 | 24.97 |
| SSIM-C | 0.9197 | 0.9198 | 0.9209 | **0.9212** | 0.0178 | 0.0088 | 0.0021 |
| SSIM-S | 0.9126 | 0.9134 | 0.9149 | **0.9158** | 0.9211 | 0.9389 | 0.9617 |
| LPIPS-C | 0.0692 | 0.0579 | 0.0486 | **0.0398** | 0.0193 | 0.0107 | 0.0042 |
| LPIPS-S | 0.0498 | 0.0541 | 0.0619 | **0.0732** | 0.0948 | 0.1112 | 0.2253 |
| NCC-C | 0.9914 | 0.9918 | 0.9924 | **0.9916** | 0.9949 | 0.9961 | 0.9984 |
| NCC-S | 0.9931 | 0.9923 | 0.9911 | **0.9901** | 0.9852 | 0.9861 | 0.9897 |

**Note:**
The bolded parts in the table indicate the cases where the overall performance of the two image pairs across all evaluation metrics reaches the optimal level.

## Comparative experiment

To validate the robustness of the DERIS model, we conducted comparative experiments with *Baluja (2017)*, RIIS (*Xu et al., 2022*), Hinet (*Jing et al., 2021*; *Weng et al., 2019*), and DKIS (*Yang, Xu & Liu, 2025*), using the DIV2K dataset under various attack conditions. Detailed results are presented in Table 5. Note that the datasets used in these comparisons are consistent with those employed by the original methods. The results indicate that DERIS achieves superior PSNR-S performance across different Gaussian noise levels and JPEG compression qualities. Under Gaussian noise with $\sigma = 1$, DERIS attains a PSNR-S of 36.91 dB, which is 11.79 dB higher than that of Hinet. Under Gaussian noise with $\sigma = 10$, DERIS achieves a PSNR-S of 30.24 dB, outperforming the second-best method by 1.08 dB. In the case of JPEG compression with a quality factor (QF) of 80, DERIS reaches a PSNR-S of 28.33 dB, exceeding Hinet by 19.32 dB. This demonstrates that Hinet exhibits poor robustness under increased noise levels. Furthermore, at a JPEG QF of 90, DERIS achieves a PSNR-S of 30.67 dB, surpassing the second-best method, PRIS, by 1.22 dB. These findings collectively indicate that DERIS possesses the highest level of robustness among the evaluated methods.

Furthermore, to verify the generalization ability of our model, we evaluate not only the model trained on the DIV2K dataset but also conduct evaluations on two additional benchmark datasets: ImageNet (ILSVRC 2017) (*Russakovsky et al., 2015*), which contains 5,500 images from the test set, and COCO (2017 Test images) (*Lin et al., 2014*), which

**Table 5 Comparison of PSNR (dB) under various attacks.** PSNR-C (left) and PSNR-S (right).

| Model | Gaussian $\sigma = 1$ | | Gaussian $\sigma = 10$ | | JPEG QF = 80 | | JPEG QF = 90 | |
|---|---|---|---|---|---|---|---|---|
| | PSNR-C | PSNR-S | PSNR-C | PSNR-S | PSNR-C | PSNR-S | PSNR-C | PSNR-S |
| *Baluja (2017)* | 21.98 | 20.67 | 21.43 | 19.78 | 13.7 | 7.52 | 14.79 | 8.22 |
| RIIS (*Xu et al., 2022*) | 29.65 | 30.29 | 27.96 | 28.04 | 27.65 | 28.21 | 28.02 | 27.99 |
| PRIS (*Yang et al., 2024*) | 38.21 | 36.86 | 31.21 | 29.16 | 29.21 | 27.91 | 31.67 | 29.45 |
| Hinet (*Jing et al., 2021*) | 38.51 | 25.12 | 36.22 | 8.36 | 34.12 | 9.01 | 34.45 | 13.67 |
| *Weng et al. (2019)* | 36.16 | 17.25 | 32.16 | 14.21 | 31.49 | 11.23 | 31.56 | 12.16 |
| DKIS (*Yang, Xu & Liu, 2025*) | 37.96 | 24.85 | 36.16 | 8.21 | 34.49 | 9.23 | 34.16 | 12.76 |
| Ours | 38.23 | 36.91 | 32.43 | 30.24 | 29.67 | 28.33 | 32.51 | 30.67 |

**Table 6 Performance of different models on various datasets under JPEG QF = 80 attack.**

| Model | Cover/Container image pair | | | | | | Secret/Extracted image pair | | | | | |
|---|---|---|---|---|---|---|---|---|---|---|---|---|
| | PSNR | SSIM | APD | RMSE | LPIPS | NCC | PSNR | SSIM | APD | RMSE | LPIPS | NCC |
| **DIV2K** | | | | | | | | | | | | |
| Baluja | 13.7 | 0.59 | 40.6 | 55.8 | 0.8007 | −0.0584 | 7.52 | 0.08 | 91.7 | 110.7 | 0.7390 | 0.2362 |
| HiNet | 34.1 | 0.93 | 3.64 | 5.24 | 0.0353 | 0.9986 | 9.01 | 0.15 | 76.1 | 91.8 | 0.8552 | 0.1363 |
| Joint | 33.4 | 0.91 | 4.12 | 5.83 | 0.0315 | 0.9906 | 8.39 | 0.11 | 83.2 | 96.4 | 0.7182 | 0.2387 |
| Weng | 30.3 | 0.87 | 5.21 | 7.69 | 0.0314 | 0.9879 | 7.46 | 0.09 | 87.6 | 106.7 | 0.7385 | 0.2369 |
| Dkis | 33.97 | 0.92 | 3.68 | 5.34 | 0.0364 | 0.9976 | 8.91 | 0.14 | 77.2 | 92.7 | 0.8659 | 0.1265 |
| RIIS | 27.6 | 0.74 | 7.32 | 11.57 | 0.2013 | 0.9801 | 28.2 | 0.83 | 7.14 | 10.21 | 0.1913 | 0.9846 |
| Pris | 29.2 | 0.81 | 6.73 | 10.46 | 0.1097 | 0.9846 | 27.9 | 0.83 | 7.21 | 10.38 | 0.1936 | 0.9813 |
| Ours | 29.7 | 0.85 | 6.34 | 9.46 | 0.0841 | 0.9946 | 28.3 | 0.84 | 6.94 | 10.01 | 0.1056 | 0.9812 |
| **ImageNet** | | | | | | | | | | | | |
| Baluja | 13.38 | 0.56 | 43 | 58.25 | 0.3999 | 0.8329 | 7.29 | 0.07 | 94 | 113.2 | 0.8374 | −0.0516 |
| HiNet | 31.97 | 0.91 | 4.51 | 6.78 | 0.0147 | 0.9970 | 9.73 | 0.13 | 77.8 | 94.5 | 0.7772 | 0.0193 |
| Joint | 32.56 | 0.92 | 4.61 | 5.93 | 0.0346 | 0.9875 | 9.13 | 0.12 | 79.3 | 96.6 | 0.7018 | 0.2378 |
| Weng | 29.84 | 0.89 | 5.92 | 8.64 | 0.0285 | 0.9763 | 10.53 | 0.14 | 65.8 | 77.3 | 0.7195 | 0.2585 |
| Dkis | 31.88 | 0.91 | 4.83 | 6.97 | 0.0249 | 0.9956 | 9.83 | 0.14 | 78.9 | 95.6 | 0.7763 | 0.0278 |
| RIIS | 26.51 | 0.69 | 8.12 | 12.64 | 0.2298 | 0.9664 | 25.41 | 0.77 | 10.1 | 11.5 | 0.2385 | 0.9585 |
| Pris | 27.59 | 0.79 | 7.11 | 10.89 | 0.1644 | 0.9888 | 25.63 | 0.79 | 9.72 | 10.89 | 0.2914 | 0.9855 |
| Ours | 28.63 | 0.84 | 6.53 | 9.32 | 0.0916 | 0.9901 | 27.74 | 0.81 | 8.31 | 10.13 | 0.1256 | 0.9711 |
| **COCO** | | | | | | | | | | | | |
| Baluja | 14.06 | 0.59 | 38 | 54.4 | 0.2892 | 0.8604 | 6.91 | 0.09 | 102 | 118.9 | 0.9184 | 0.1808 |
| HiNet | 33.37 | 0.91 | 4.1 | 5.8 | 0.0138 | 0.9932 | 8.72 | 0.11 | 78 | 95.1 | 0.9240 | 0.1122 |
| Joint | 34.13 | 0.93 | 5.7 | 7.7 | 0.0249 | 0.9916 | 12.04 | 0.21 | 65 | 78.9 | 0.9163 | 0.3244 |
| Weng | 31.04 | 0.88 | 5.9 | 7.3 | 0.0344 | 0.9907 | 10.39 | 0.14 | 69 | 89.3 | 0.7514 | 0.0369 |
| Dkis | 33.21 | 0.91 | 4.6 | 6.1 | 0.0149 | 0.9912 | 8.99 | 0.12 | 76 | 94.3 | 0.9211 | 0.1323 |
| RIIS | 27.2 | 0.71 | 7.1 | 11.2 | 0.2118 | 0.9785 | 25.8 | 0.78 | 9.6 | 13.7 | 0.2317 | 0.9645 |
| Pris | 28.07 | 0.77 | 6.8 | 10.4 | 0.0285 | 0.9920 | 25.9 | 0.78 | 9.5 | 13.5 | 0.1475 | 0.9694 |
| Ours | 28.44 | 0.81 | 6.9 | 10.7 | 0.0961 | 0.9831 | 26.81 | 0.79 | 9.3 | 11.7 | 0.2151 | 0.9511 |

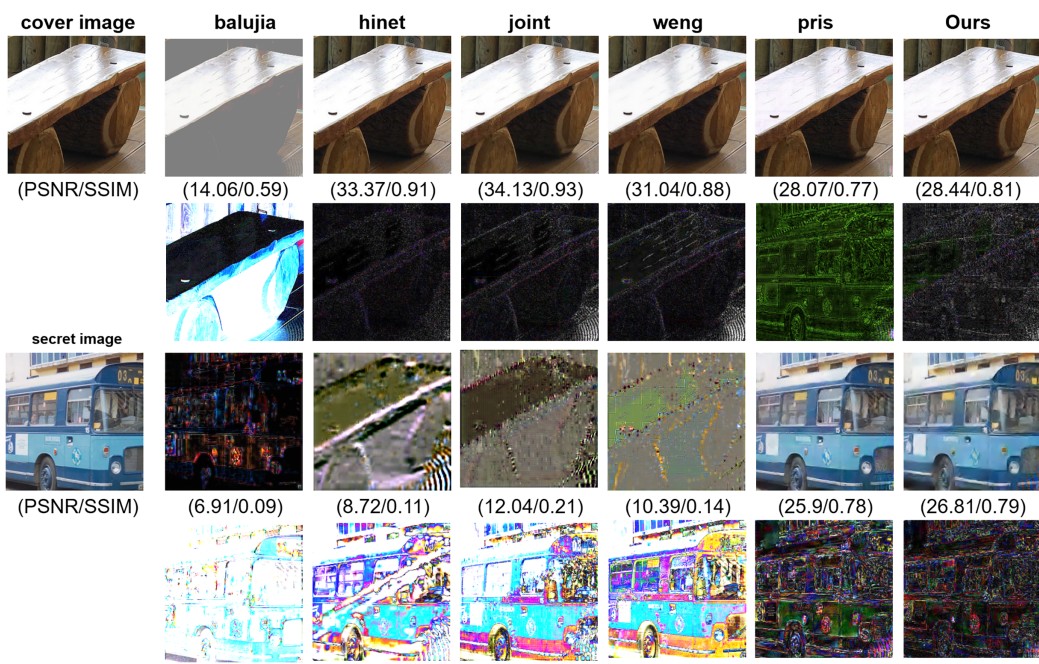

**Figure 7 The visualization results of DERIS and other models under JPEG QF = 80 attack.**

contains 400 images from the test set. As shown in Table 6, under JPEG compression with a quality factor (QF) of 80, the table presents a performance comparison of DERIS against existing methods, including *Baluja (2017)*, Hinet (*Jing et al., 2021*), Joint (*Zhang et al., 2023*; *Weng et al., 2019*), DKIS (*Yang, Xu & Liu, 2025*), RIIS (*Xu et al., 2022*), and PRIS (*Yang et al., 2024*), across different datasets. It is important to note that the datasets used in these comparisons are consistent with those employed by the original methods. Compared to other state-of-the-art methods, our DERIS model demonstrates excellent performance on the cover-container image pairs and achieves the best results on the secret-extracted image pairs. On the DIV2K dataset, DERIS achieves a PSNR-S of 28.3 dB, outperforming the second-best method, PRIS, by 0.4 dB. On the ImageNet dataset, DERIS attains a PSNR-S of 27.74 dB, exceeding PRIS by 2.11 dB. Similarly, on the COCO dataset, DERIS reaches a PSNR-S of 26.81 dB, surpassing PRIS by 0.91 dB. These results indicate that DERIS exhibits superior robustness and is capable of withstanding various types of attacks.

As shown in Fig. 7, the visualization results of different models using the COCO dataset under JPEG compression with quality factor (QF) = 80 are presented. The second row displays the residual maps of the cover-container image pairs, while the fourth row shows the residual maps of the secret-extracted image pairs. In the first two rows of the cover-container image pairs, all models except Baluja exhibit excellent visual performance. In the third row, which shows the visual results of the secret-extracted image pairs, only RIIS, PRIS, and our model successfully recover the detailed appearance of the original secret image. This indicates that the other models demonstrate inferior robustness under this attack condition. Our model achieves a PSNR-S that is 0.91 dB higher than both RIIS and PRIS. In addition, the PSNR of our container image is 1.37 dB higher

than that of RIIS's container image and 0.34 dB higher than that of PRIS's container image. In the fourth row of residual maps, the residual maps of the first four models still reveal visible traces of the original image, indicating suboptimal steganographic performance. The residual maps of RIIS and PRIS exhibit more complex textures and features, with RIIS showing more severe artifacts compared to PRIS, suggesting that the embedded signals or image residuals are not fully concealed. In contrast, our residual map exhibits a relatively uniform noise distribution, indicating that the embedded signal causes less distortion to the original image. This leads to better steganographic performance and makes the modifications harder to detect by the naked eye or through simple statistical analysis.

## Security analysis

In the task of image steganography, ensuring the security of stego-images is of paramount importance. Highly secure stego-images are essential in scenarios where covert information transmission must be concealed, such as in government communications, intelligence transfer, or the protection of sensitive corporate data. To comprehensively evaluate the detection-resistance capabilities of stego content generated by various image-hiding methods, this section employs four mainstream deep learning-based steganalysis models—Yedroudj-Net (*Yedroudj, Comby & Chaumont, 2018*), Xu-Net (*Xu et al., 2025*), StegNet (*Wu et al., 2018*), and ZhuNet (*Zhu et al., 2020*)—as measurement tools to assess the security of steganographic methods. These models are widely used in the field of steganalysis and can quantify the detectability of stego-images, thereby providing an objective assessment of the security performance of different steganographic methods. Yedroudj-Net is a convolutional neural network specifically designed for deep learning-based image steganalysis. It can capture subtle noise patterns in images and evaluate the detectability of stego content. The network consists of multiple convolutional layers and residual connections. It takes an image as input and outputs a binary classification result by learning the steganographic features in the image, such as noise residuals. Xu-Net is another high-performance convolutional network that focuses on extracting high-frequency features from images, making it particularly suitable for detecting low-capacity stego content. It first applies high-pass filtering to preprocess the input image, then uses multiple convolutional layers to learn feature representations, and finally classifies the image through a fully connected layer. StegNet combines convolutional neural networks with an autoencoder architecture, providing stable detection performance across different steganographic methods. It is suitable for evaluating the anti-detection capability of various deep learning-based steganographic approaches. The network takes stego-images as input, extracts features through the encoder, reconstructs the feature space *via* the decoder, and finally performs binary classification through a discriminator to determine whether an image is stego. ZhuNet is currently recognized as one of the most powerful deep learning-based steganalysis networks, particularly effective in complex steganographic scenarios, and serves as an important benchmark for assessing the robustness of steganographic methods. It uses deep convolutional layers and multi-scale feature fusion to extract fine-grained image features and outputs the steganography

**Table 7  Accuracy of different steganalysis models.**

| Methods | Yedroudj-Net | Xu-Net | Steg-Net | Zhu-Net |
|---|---|---|---|---|
| *Baluja (2017)* | 82 ± 2 (%) | 83 ± 2 (%) | 84 ± 2 (%) | 81 ± 3 (%) |
| HiNet (*Jing et al., 2021*) | 85 ± 2 (%) | 83 ± 2 (%) | 89 ± 2 (%) | 79 ± 2 (%) |
| Joint (*Zhang et al., 2023*) | 97 ± 2 (%) | 96 ± 2 (%) | 98 ± 2 (%) | 98 ± 2 (%) |
| *Weng et al. (2019)* | 82 ± 2 (%) | 84 ± 2 (%) | 89 ± 2 (%) | 80 ± 3 (%) |
| PRIS (*Yang et al., 2024*) | 94 ± 2 (%) | 95 ± 2 (%) | 97 ± 2 (%) | 96 ± 2 (%) |
| Dkis (*Yang, Xu & Liu, 2025*) | 93 ± 2 (%) | 94 ± 2 (%) | 97 ± 2 (%) | 96 ± 2 (%) |
| StegFormer (*Ke, Wu & Guo, 2024*) | 81 ± 2 (%) | 79 ± 2 (%) | 83 ± 2 (%) | 72 ± 2 (%) |
| Ours | 84 ± 2 (%) | 82 ± 2 (%) | 85 ± 2 (%) | 80 ± 2 (%) |

detection result through a fully connected classifier. These steganalysis models, as measurement tools, provide a quantitative evaluation of steganographic methods' security. By using their detection rate metrics, we can assess the concealment and robustness of steganographic methods against advanced analysis techniques, ensuring that the proposed methods maintain security in practical applications. For our experiments, we randomly selected 400 images from the COCO dataset, with 200 designated as cover images and the remaining 200 used as secret images. Using various steganographic network models, we embedded the secret images into the cover images, resulting in 200 stego-images. Subsequently, we selected 100 of these stego-images as samples simulating those identified by attackers and used them to train the four steganalysis models: Yedroudj-Net, Xu-Net, StegNet, and ZhuNet. Finally, the remaining 100 stego-images were tested using the trained models to evaluate the detection rates of different steganographic methods. As shown in Table 7, our method demonstrates strong concealment across four mainstream deep learning-based steganalysis networks. Specifically, the detection rates on Yedroudj-Net and Xu-Net are 84 ± 2% and 82 ± 2%, respectively, which are only 3 ± 2% higher than those of the current state-of-the-art method, StegFormer. On StegNet, our method achieves a detection rate of 85 ± 2%, representing a 2 ± 2% improvement over StegFormer. Notably, on ZhuNet—which exhibits the strongest detection capability among the evaluated models—our method performs the best, achieving a detection rate of 80 ± 2%. These results clearly demonstrate that our proposed method exhibits strong robustness against advanced deep learning-based steganalysis models.

## Performance validation under distortion-free conditions

To demonstrate that our model maintains favorable performance in the absence of image distortions, we evaluated DERIS on 400 images from the COCO test dataset and compared its performance with several state-of-the-art steganographic models. The detailed results are summarized in Table 8. As can be observed, DERIS achieves competitive performance even under no-interference conditions. Specifically, the PSNR of the Cover-Steg image pairs reaches 42.11 dB, which is only 2.52 dB lower than the highest result achieved by StegFormer. Moreover, the PSNR of the Secret-Extracted image pairs reaches 46.41 dB, showing a gap of merely 2.44 dB behind StegFormer.

**Table 8 Performance of different models on the COCO dataset under no interference conditions.**

| Model | Cover/Container image pair | | | | | | Secret/Extracted image pair | | | | | |
|---|---|---|---|---|---|---|---|---|---|---|---|---|
| | PSNR | SSIM | APD | RMSE | LPIPS | NCC | PSNR | SSIM | APD | RMSE | LPIPS | NCC |
| *Baluja (2017)* | 35.46 | 0.921 | 3.29 | 4.49 | 0.0117 | 0.9961 | 32.46 | 0.949 | 4.99 | 6.27 | 0.0025 | 0.9956 |
| Hinet (*Jing et al., 2021*) | 42.67 | 0.985 | 1.34 | 2.03 | 0.0015 | 0.9985 | 48.22 | 0.993 | 0.77 | 1.27 | 0.0003 | 0.9989 |
| Joint (*Zhang et al., 2023*) | 28.79 | 0.939 | 11.17 | 13.07 | 0.0179 | 0.9931 | 30.81 | 0.930 | 7.09 | 9.67 | 0.0533 | 0.9845 |
| *Weng et al. (2019)* | 30.84 | 0.917 | 6.80 | 8.34 | 0.0159 | 0.9945 | 35.58 | 0.965 | 3.53 | 4.67 | 0.0309 | 0.9938 |
| Pris (*Yang et al., 2024*) | 38.77 | 0.967 | 2.18 | 2.98 | 0.0052 | 0.9979 | 38.91 | 0.981 | 2.20 | 3.09 | 0.0033 | 0.9981 |
| Dkis (*Yang, Xu & Liu, 2025*) | 34.67 | 0.921 | 3.48 | 4.99 | 0.0093 | 0.9950 | 30.05 | 0.913 | 5.47 | 8.67 | 0.0232 | 0.9864 |
| StegFormer (*Ke, Wu & Guo, 2024*) | 44.63 | 0.991 | 0.97 | 1.68 | 0.0005 | 0.9995 | 48.85 | 0.9994 | 0.725 | 1.20 | 0.0005 | 0.9994 |
| Ours | 42.11 | 0.981 | 1.43 | 2.11 | 0.0021 | 0.9976 | 46.41 | 0.991 | 0.83 | 1.31 | 0.0009 | 0.9979 |

## Steganographic key necessity verification experiment

In practical applications, attacks can occur during both the embedding and extraction stages. An attack on the embedding process involves the unauthorized insertion of a forged secret image into a host image. The attacker attempts to generate a deceptive container image by manipulating the host image and a fake secret image. The success of this attack depends on whether the target steganographic method mistakenly extracts the forged secret from the resulting container image. On the other hand, an attack on the extraction process aims to bypass the legitimate extraction mechanism in order to illegally recover the secret image, even in the absence of the correct private key or through the use of incorrect extraction methods. Based on our proposed model, we designed an attack simulation network to evaluate the security of both the embedding and extraction processes. In the embedding attack scenario, the simulation network was trained using the host image and the secret image as inputs, and the container image as the output, aiming to mimic the behavior of the genuine embedding process. For the extraction attack scenario, the network was trained using the container image as input and the secret image as output, attempting to bypass the original extraction procedure and directly reconstruct the secret image. The experiments were conducted on a test set of 5,500 images from the ImageNet dataset, with JPEG compression (quality factor, QF = 80) applied uniformly as the attack condition. As shown in Fig. 8: the first column displays the original secret images; the second and third columns present the results of attacks on the embedding process, where the second column corresponds to the case without a key (PSNR = 27.53 dB), and the third column shows the case with a key, where the PSNR drops significantly to 8.32 dB. The fourth and fifth columns illustrate the results of attacks on the extraction process, with PSNR values of 27.74 dB (without a key) and 11.35 dB (with a key), respectively. Visually, it is also evident that the attack performance is significantly better when the correct key is not used. In conclusion, the experimental results highlight the critical role of the cryptographic key in enhancing system security. In both embedding and extraction attack scenarios, an attacker cannot effectively recover the secret information without the correct key, demonstrating the effectiveness of our method in ensuring information security.

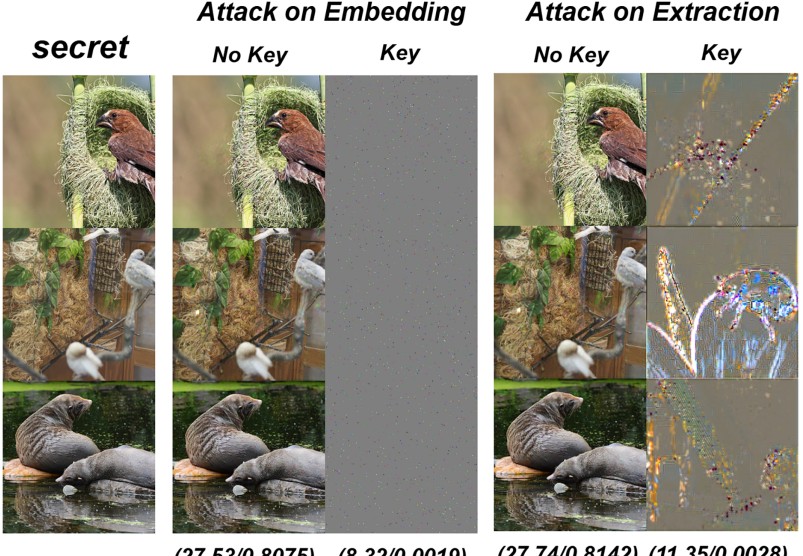

**Figure 8** Steganographic key necessity verification experiment.

# CONCLUSION

In this article, we propose a novel Denoising and Enhancement Robust Image Steganography (DERIS) model based on reversible neural networks. By introducing two identical denoising and enhancement modules before the DWT in the reverse extraction process and after the IDWT, along with a corresponding "denoising and enhancement training strategy," DERIS significantly improves the robustness of the image steganography system and enhances the quality of the extracted images. Experimental results demonstrate that, compared to state-of-the-art methods, DERIS achieves more stable recovery of high-quality extracted images under various distortions such as Gaussian noise, Poisson noise, and lossy compression. Specifically, DERIS exhibits significant advantages in several key metrics including PSNR, SSIM, APD, RMSE, LPIPS and NCC. Future work will focus on further optimizing algorithm efficiency to enable adaptive noise level detection, thereby determining whether to activate the denoising and enhancement modules. Additionally, we aim to explore the potential applications of DERIS in broader scenarios, expanding its utility across diverse domains.

## Funding

This work was supported by the National Natural Science Foundation of China (62176140). The funders had no role in study design, data collection and analysis, decision to publish, or preparation of the manuscript.

## Grant Disclosures

The following grant information was disclosed by the authors:
National Natural Science Foundation of China: 62176140.

## Competing Interests

The authors declare that they have no competing interests.

## Author Contributions

- Dapeng Cheng conceived and designed the experiments, authored or reviewed drafts of the article, and approved the final draft.
- Yue Kong conceived and designed the experiments, performed the experiments, analyzed the data, performed the computation work, prepared figures and/or tables, authored or reviewed drafts of the article, and approved the final draft.
- Yanyan Mao analyzed the data, prepared figures and/or tables, and approved the final draft.
- Liunian Bian analyzed the data, prepared figures and/or tables, and approved the final draft.

## Data Availability

The code is available in the Supplemental File.

The DIV2K dataset is available at: Computer Vision Lab (CVL) at ETH Zurich. https://data.vision.ee.ethz.ch/cvl/DIV2K/.

The COCO (Common Objects in Context) dataset is available at the COCO Consortium: https://cocodataset.org/#download.

The ImageNet dataset is available through Princeton University and Stanford University: https://www.image-net.org/challenges/LSVRC/index.php.

## Supplemental Information

Supplemental information for this article can be found online at http://dx.doi.org/10.7717/peerj-cs.3368#supplemental-information.

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
