# Peer review of "Denoising and enhancement for robust image steganography"

_PeerJ Computer Science, doi:10.7717/peerj-cs.3368_

## Round 0.1 · original submission · Major Revisions

· Academic Editor

Major Revisions

Reviewer 1 ·

Basic reporting

The topic is timely and significant; however, the manuscript would benefit from improvements in the following areas:

Abstract: Strengthen the abstract by clearly presenting the main findings. Specifically, include the best results obtained, the evaluation metrics and datasets used, and the improvements achieved compared to existing methods.

Introduction: The introduction relies heavily on outdated references. It should be updated with recent studies and reviews. The authors are encouraged to consult the following relevant works:

For image steganography: Data hiding and its applications: Digital watermarking and steganography

For LSB and Invertible CNN techniques: High-capacity speech steganography for the G723.1 coder based on quantised line spectral pairs interpolation and CNN auto-encoding

Related Work: The related work section lacks a discussion on recent image denoising approaches. A dedicated subsection is recommended to review state-of-the-art image denoising methods, such as:

Reinforced Residual Encoder–Decoder Network for Image Denoising via Deeper Encoding and Balanced Skip Connections

Experimental design

- Der Module lacks of mathematical description

- Exchanged Steganographic key is missing

Validity of the findings

Security Analysis: The manuscript lacks a subsection discussing the robustness of the proposed method against recent deep learning-based steganalysis techniques (e.g., Yedroudj-Net, Xu-Net). The following reference provides a useful taxonomy and reproducing tools:

Deep learning for steganalysis of diverse data types: A review of methods, taxonomy, challenges, and future directions

In addition, the authors should include a comparative table evaluating the robustness of the proposed method against existing steganalysis approaches, using the same datasets to ensure a fair comparison.

Additional comments

Figures: Several figures, especially Figure 1, are blurry and difficult to read. Please improve the quality and resolution of all figures to meet publication standards.

Cite this review as

Reviewer 2 ·

Basic reporting

see additional comment

Experimental design

see additional comment

Validity of the findings

see additional comment

Additional comments

This paper proposes an INN-based DERIS model, which aims to improve robustness against noise and compression attacks in image steganography systems. Although the INN architecture has been used before, the main contribution of this paper is the addition of two denoising modules and a stepwise training strategy. Some substantial comments need to be noted:
1. This model explicitly emphasizes robustness against disturbances (noise, JPEG compression), not evaluating the undetectability aspect. This is more in line with the purpose of watermarking, rather than classical steganography which emphasizes imperceptibility and undetectability.
2. If it wants to be more steganographic, this paper should include steganalysis testing.
3. The evaluation is limited to PSNR, SSIM, RMSE, and MAE, without considering modern metrics such as LPIPS and FID, regarding this, it is necessary to add a reference https://doi.org/10.62411/faith.3048-3719-76

4. The DERIS architecture always activates the denoising module, even when the container image is not disturbed. This poses a risk of overprocessing and can actually harm the extraction performance in ideal scenarios. The design should be adaptive to noise levels or input conditions.
5. The ablation study is limited to a combination of denoising modules and training strategies, but does not evaluate the sensitivity to model parameter variations.
6. Although comparisons are made with several related methods, it is not explained whether the baselines were retrained on the same dataset. This could potentially create a gap in the validity of the “state-of-the-art” claim.
7. Although the practical motivations (noise, compression, transmission) are quite clear, the paper does not elaborate in depth on the research gaps of previous models, so the methodological contribution of DERIS is not fully substantiated academically.

Cite this review as

·

Basic reporting

• In-text citations often appear clumped together without proper punctuation or spacing, e.g., “(Xu et al., 2022) (Yang et al., 2024)”. Ensure all references are properly cited
• Figures are dense, and also small fonts are used. These should be improved for better legibility
• The Related Work section is thorough but would be much clearer with a summarizing table comparing key models, their techniques, strengths, and limitations. This would strengthen the rationale for proposing DERIS.
• The manuscript should be proofread for grammatical errors

Experimental design

• The experiments utilize multiple publicly available image datasets — DIV2K, COCO, and ImageNet — which ensures a broader generalizability of the results and demonstrates an effort to validate robustness across contexts. The inclusion of detailed ablation experiments helps justify the design choices
• The methodology is solid in concept, but technical descriptions lack sufficient depth for replication. Key elements such as hyperparameter settings, loss weight choices, and structural configurations of modules are not adequately disclosed
• The manuscript does not address the training time, model size, or inference speed.

Validity of the findings

The DERIS model consistently outperforms baseline methods across multiple standard metrics (PSNR, SSIM, RMSE, and APD) and under various attack conditions such as Gaussian noise and JPEG compression. By evaluating on DIV2K, COCO, and ImageNet, the study shows that DERIS maintains performance across diverse image distributions, lending credibility to its claims of generalization and robustness.

Additional comments

The paper identifies a critical shortfall in current image steganography methods: poor robustness under typical image degradations such as Gaussian noise and lossy compression. The study justifies its investigation by highlighting how the accuracy of secret image retrieval drastically drops when container images are distorted. This is a valid and significant concern, especially in security-sensitive applications like medical imaging, military communications, and digital rights management.

The paper will be strengthened with the comments given above

Cite this review as

---

## Round 0.2 · Major Revisions

· Academic Editor

Major Revisions

**Language Note:** When preparing your next revision, please ensure that your manuscript is reviewed either by a colleague who is proficient in English and familiar with the subject matter, or by a professional editing service. PeerJ offers language editing services; if you are interested, you may contact us at [email protected] for pricing details. Kindly include your manuscript number and title in your inquiry. – PeerJ Staff

Reviewer 2 ·

Basic reporting

-

Experimental design

-

Validity of the findings

-

Additional comments

This paper has undergone several revisions, but after rereading it, I still find several things that need improvement and new points that need to be addressed:

1. A measurement tool has been added, but there is no robust analysis. There is no explanation of why this measurement tool is important and how it is implemented. It is simply mentioned without a meaningful explanation.

2. The reason for using images as messages needs to be explained. Steganography also uses text as a message. If the reason is to make it stronger, how is it different from watermarking? The focus of steganography needs to be clarified. See the paper "Digital Image Steganography Survey and Investigation (Goal, Assessment, Method, Development, and Dataset)" for the direction of steganography research.

3. The results presented in Tables 2, 3, and 4 only focus on tests with Gaussian noise attacks.

4. I haven't found any results without Gaussian noise attacks.

5. In general, Gaussian noise attacks are stronger than JPEG compression. Why are the results shown in Table 5 the opposite?

6. Comparison tables like Tables 5 and 6 need clarification: do they use the same dataset as the method replicated, or are they simply taken from the proposed paper?

7. There is no explanation regarding the payload.

Cite this review as

---

## Round 0.3 · accepted · Accept

· Academic Editor

Accept

The authors efficiently addressed the points raised by the reviewers and therefore I can recommend this article for acceptance.

Reviewer 2 ·

Basic reporting

After the revision, now is better.

Experimental design

-

Validity of the findings

-

Cite this review as